# Immunotherapy for Uterine Cervical Cancer Using Checkpoint Inhibitors: Future Directions

**DOI:** 10.3390/ijms21072335

**Published:** 2020-03-27

**Authors:** Masahiro Kagabu, Takayuki Nagasawa, Chie Sato, Yasuko Fukagawa, Hanae Kawamura, Hidetoshi Tomabechi, Shuji Takemoto, Tadahiro Shoji, Tsukasa Baba

**Affiliations:** Department of Obstetrics and Gynecology, Iwate Medical University School of Medicine, Morioka, Iwate 028-3695, Japan; tnagasaw@iwate-med.ac.jp (T.N.); c.kurokawa0825@gmail.com (C.S.); n5mykkhanae@yahoo.co.jp (H.K.); suga_yasuko@yahoo.co.jp (Y.F.); bechitomabehi@gmail.com (H.T.); tshyuuji@gmail.com (S.T.); tshoji@iwate-med.ac.jp (T.S.); babatsu@iwate-med.ac.jp (T.B.)

**Keywords:** cervical cancer, immunotherapy, immune checkpoint inhibitor, chemoradiotherapy

## Abstract

Immune checkpoint inhibitors (ICIs) have demonstrated marked clinical effects worldwide, and “cancer immunotherapy” has been recognized as a feasible option for cancer treatment. Significant treatment responses have already been attained for malignant melanoma and lung cancer, ahead of gynecologic cancer. In cervical cancer, however, results are only available from phase II trials, not from phase III trials. Cervical cancer is a malignant tumor and is the fourth most common cancer among women worldwide. Since the introduction of angiogenesis inhibitors, treatment for recurrent and advanced cervical cancers has improved in the past five years, but median overall survival is 16.8 months for advanced cervical cancer, and all-stage five-year overall survival rate is 68%, indicating that treatment effects remain inadequate. For this reason, the development of new therapeutic approaches is imperative. We describe herein the KEYNOTE-158 and CheckMate 358 clinical trials, which were conducted for cervical cancer, and discuss future directions, including potential combinations with concurrent chemoradiation therapy (CCRT), as noted for other types of cancer.

## 1. Introduction

Cancer immunotherapy is a general term for treatments that strengthen or trigger the immune system of the patient to elicit antitumor effects. Cancer immunotherapy has a long history, starting with peptide vaccines and adoptive immunotherapy for cancer, although promising outcomes have remained elusive. However, with the recent introduction of immune checkpoint inhibitors (ICIs) that release immune-suppressing brakes, evidence for the feasibility of immunotherapy in cancer treatment has finally become established and is revolutionizing cancer treatment. Typically, dendritic cells engulf cancer cells, migrate to the lymph nodes and present cancer antigens to T cells in lymph nodes. T cells subsequently recognize cancer, but an immune co-signal that determines the T-cell function is necessary at this point [1,2]. This co-signal can be promotional or suppressive in nature. In the promotion type, T cells become activated, whereas in the suppression type, T cells recognize but cannot attack the cancer cells. Furthermore, in the cancer microenvironment, cancer cells express suppression signals on the cell surface to suppress the immune function of T cells activated in lymph nodes. Lymph nodes and the cancer microenvironment possess separate mechanisms to determine the directionality of immune response to cancer cells. These are known as immune checkpoints [3]. Cervical cancer is the fourth most common cancer among women, and the seventh most common of all human cancers. Approximately 569,000 cases of cervical cancer and 311,000 deaths from disease occurred in 2018, according to the International Agency for Research on Cancer database [4]. Primary treatment for early stage cervical cancer is surgery. Radio- and chemotherapy have been used to treat patients with advanced uterine cervical cancer [5], but these therapies have met with limited success. Overall survival (OS) of the GOG204 study was extended from 13 to 17 months in recurrent and unresectable cervical cancer, with the addition of bevacizumab to platinum-based chemotherapy [6]. However, because regimens for second-line treatment and beyond have yet to be established, further development of new methods is anticipated.

Against this background, ICIs have become highly anticipated as a potential treatment for cervical cancer. Human papilloma virus (HPV) is known to be involved in the development of cervical cancer. Consequently, immunotherapy targeting HPV has been attempted, but insufficient effects were obtained [7]. For this reason, clinical trials have been conducted to assess the efficacy of ICIs against cervical cancer. In particular, pembrolizumab was shown to be effective in solid tumors, including cervical cancer, according to the KEYNOTE-158 clinical trial [8], and was approved by the Food and Drug Administration (FDA) in the United States. In this article, we provide an overview of the tumor immune environment in cancer tissues and programmed death 1 (PD1)/programmed death ligand 1 (PD-L1) expression in cervical cancer tissues, and we describe results from the KEYNOTE-158 trial and the latest CheckMate 358 trial. We also discuss the potential combination of ICIs and radiation therapy, which is known to strengthen the immune system.

## 2. Cervical Cancer Immunopathogenesis Associated with Effects of ICIs

The immunopathogenesis of cancer varies, but there are two main types of cancer in which the main effectors are CD8+ T cells specific to tumor antigens (i.e., neoantigens) can be divided into “T-cell-inflamed”, where antitumor T cells aggregate before treatment, and “non-T-cell-inflamed”, where such aggregation is absent. Cytokines such as interferon (IFN)-γ, which are secreted by tumor antigen-specific T cells that recognize cancer cells, induce cancer cells and surrounding macrophages to express PD-L1, thereby suppressing CD8+ T cells and cancer elimination in T-cell-inflamed cancer [9]. Tumor antigen-specific T cells require the presence of highly immunogenic tumor antigens such as cancer/testis antigens and neoantigens as targets. T-cell-target antigens are also abundant in cancer that presents multiple DNA mutations, such as ultraviolet-radiation-induced malignant melanoma, smoking-related lung cancer, cancer with abnormalities in DNA repair genes, such as BRCA1/2, and cancer with high microsatellite instability, likely resulting in a T-cell-inflamed environment. Moreover, prolonged infection by HPV is known to be profoundly associated with the carcinogenesis of cervical cancer, and significantly affects PD-L1 expression in tumor tissue. This is also considered T-cell-inflamed cancer. PD-L1 expression on the cell membrane and IFN-γ mRNA upregulation were observed in HPV-related head and neck squamous cell carcinoma (SCC). This result exhibited that IFN-γ was secreted through initial HPV infection, and subsequently induced PD-L1 expression [10]. Several teams investigated whether HPV infection affects PD-L1 expression in cervical cancer, revealing that HPV-positive status correlated positively with increased PD-L1 expression [11,12]. PD-L1 is expressed on the surface of cervical cancer tumor cells, antigen-presenting cells and tumor-infiltrating lymphocytes (TILs), whereas most PD-1-positive cells have been identified as T cells in the stroma of cervical cancer. PD-1 expression in the tumor stroma of cervical cancer is observed in 46.9%~60.8% of patients [13,14]. Various researchers have also investigated PD-L1 expression in cervical cancer tissue. PD-L1 expression is observed in 34.4%–96% of cervical cancer tissues, but is rarely observed in histologically normal cervical tissue [15,16]. Analysis by histology type observed PD-L1 expression in 80% of cervical SCC [12]. PD-L1 amplification or acquisition was observed in 22% of patients with cervical SCC in the TCGA database [17]. Furthermore, PD-L1 can be expressed on TILs, and this plays a role in the antitumor-response blockade. PD-L1 expression rates in cancer cells and TILs were 59.1% and 47.0% in a study of cervical SCC samples [14]. These data suggest that both PD-L1 and PD-1 are widely expressed in cervical cancer tumor cells and stroma, indicating potential treatment targets for PD-1/PD-L1 inhibitors.

On the other hand, because some types of cervical cancer are unrelated to HPV, the existence of non-T-cell-inflamed cancer needs to be considered. As a mechanism for this, when no highly immunogenic tumor antigens, such as neoantigens, are present, the function of the molecules and cells required for antitumor T-cell induction or an immunosuppressive mechanism becomes deficient or decreased. For example, non-smoker lung adenocarcinoma that presents with super-driver mutations, such as epidermal growth factor receptor mutation, displays few DNA mutations, and antitumor T cells are less likely to be induced due to the activation of immunosuppressive pathways through cancer gene activation and signal activation. Moreover, in malignant melanoma, non-T-cell-inflamed cancer ensues due to the occurrence of multiple immunosuppressive mechanisms, such as decreases in the chemokines that recruit the dendritic cells required for T-cell induction through the activation of β-catenin and AKT signals, and the production of immunosuppressive molecules, such as vascular endothelial growth factor [18]. Moreover, in alterations of chromosome/arm somatic copy numbers such as aneuploidy, non-T-cell-inflamed cancer also likely appears, although the mechanisms remain unclear. When tumor antigens are present even in such a non-T-cell-inflamed environment, with immune manipulation that augments antitumor T-cell induction (such as adjuvant and cytokines) and removal of negative factors by eliminating and blocking immunosuppressive molecules and cells, PD-1/PD-L1 inhibition may become effective.

## 3. Evidence from Clinical Research for Cervical Cancer with ICIs

Clinical trials on various ICIs have been conducted for cervical cancer since 2015. A clinical trial of ICIs in cervical cancer that should be noted is KEYNOTE-158, which investigated pembrolizumab. Based on the study results, the FDA approved the use of pembrolizumab for cervical cancer. CheckMate 358 investigated nivolumab, a PD1-blocking antibody similar to pembrolizumab, in a similar patient population, and is another study that should be noted. The KEYNOTE-028 (phase Ib) and KEYNOTE-158 (phase II) studies investigated pembrolizumab in recurrent and unresectable cervical cancers. In KEYNOTE-028, pembrolizumab was given at 10 mg/kg, every two weeks. Twenty-four patients participated, and the objective response rate (ORR) was 17%; six-month progression-free survival (PFS) was 13%, and six-month OS was 66.7% [19]. Based on these results, KEYNOTE-158 was conducted as a phase II basket study, investigating the antitumor activity and safety of pembrolizumab in multiple cancer types. These are interim results from patients with previously treated advanced cervical cancer. Ninety-eight patients were enrolled, including 97 patients with stage III or IV disease. Eighty-two patients (83.7%) showed PD-L1-positive tumors, and 85 (86.7%) had previously received radiotherapy. Pembrolizumab (200 mg) was administered intravenously over 30 min, every three weeks, for up to two years. Patients with stable disease who discontinued treatment and exhibited subsequent disease progression were eligible for an additional one year of pembrolizumab. The ORR was 12.2% (12 patients). Eleven of the patients who responded had International Federation of Gynecology and Obstetrics (FIGO) stage IVB, disease and one had FIGO stage IIIB. All 12 responses were in patients with PD-L1-positive tumors, including one patient with adenocarcinoma. The ORR was 14.6% in the population of patients with PD-L1-positive tumors, compared to 14.3% in those who had previously received one or more lines of chemotherapy for recurrent or metastatic disease. The median PFS was 2.1 months (95% confidence interval (CI), 2.0–2.2 months) in the total population. In the PD-L1-positive tumor population, the median PFS was 2.1 months (95% CI, 2.1–2.3 months). The median OS was 9.4 months (95% CI, 7.7–13.1 months) in the total population and 11 months (95% CI, 9.1–14.1 months) in the PD-L1-positive tumor population [8].

CheckMate 358 was a phase I/II basket study investigating the antitumor activity and safety of nivolumab for virus-associated tumors. The study provided interim results from patients with previously treated recurrent or metastatic cervical cancer. Nineteen patients with cervical cancer were enrolled. Eighteen patients were FIGO stage III or IV. Seventeen of the 18 patients (89.5%) had previously received radiotherapy. Nivolumab 240 mg was administered intravenously every two weeks, for up to two years. The ORR was 26.3% (five patients). Three patients had prior surgery and radiotherapy, and two of those three patients had received prior systemic therapies for recurrent/metastatic disease. In the population of patients with PD-L1-positive tumors, the ORR was 20.0%. The median PFS was 5.1 months (95% CI, 1.9–9.1 months) in the total population. The median OS was 21.9 months (95% CI, 15.1 months to not reached) in the total population and 19.9 months (95% CI, 1.3 months to not reached) in the PD-L1-positive tumor population [20].

These two studies demonstrated that pembrolizumab and nivolumab are likely to prove useful against advanced and recurrent cervical cancer. With pembrolizumab, PD-L1 expression was shown to be a potential marker for predicting response. However, PD-L1-positive expression was also shown to not necessarily promise an effect. Moreover, with nivolumab, the effects and PD-L1 expression were not correlated. Upon first glance, results from CheckMate 358 seem to suggest the superiority of nivolumab over pembrolizumab for advanced recurrent cervical cancer. However, several issues with CheckMate 358 must be considered. First, the number of enrolled patients was low. Second, reductions in tumor size were measured by the attending physician rather than by central review. To properly evaluate nivolumab for advanced recurrent cervical cancer, a study with a greater sample size needs to be conducted, with a central review. Unfortunately, the only phase III trials investigating immune checkpoint inhibitors in cervical cancer are currently ongoing, whereas other cancer types have already obtained established data from phase III trials, and data have yet to emerge. However, there were no clinical trials that directly compared nivolumab and pembrolizumab in other cancer types. Therefore, it is not able to reveal the clinical difference between nivolumab and pembrolizumab. However, we can identify some clinical differences by comparing the result of the clinical trial that an examination design closely resembles. From results of phase III clinical trials, pembrolizumab is more effective than nivolumab in the primary treatment of stage IV non-small-cell lung cancer, and nivolumab is more effective than pembrolizumab in the second treatment of stage IV head and neck cancer. Nivolumab is more effective than pembrolizumab in the treatment after standard treatment of stage IV gastric cancer, and pembrolizumab is more effective than nivolumab in the second treatment of stage IV urothelial cancer. We await the results of phase III trials of cervical cancer (Table 1 and Table 2).

## 4. Future Directions of Immunotherapy for Cervical Cancer

Previous studies on ICIs in cervical cancer have applied a monotherapy approach. Ongoing or planned ICI studies in cervical cancer are applying combination therapy, with the aim of achieving greater response rates. Specifically, these studies are investigating combinations with existing methods (radio- or chemotherapy) or combination therapy with other molecularly targeted drugs [7].

### 4.1. Chemotherapy with ICIs for Cervical Cancer

Phase III trials with ICIs in cervical cancer are constructed from combination therapies aimed at achieving higher response rates. These studies examine a combination with existing methods (radiation therapy or chemotherapy) or combination therapy with other molecularly targeted drugs. These trials are investigating the addition of ICIs, to systemic chemotherapy (paclitaxel/carboplatin or cisplatin with bevacizumab) in unresectable or recurrent advanced cervical cancer. Platinum-based chemotherapy can lead to immunogenic cell death alone [21]. On the other hand, combinations of anti-angiogenic agents and immune checkpoint inhibitors are expected. Hypoxia contributes to immune suppression by activating Hypoxia-inducible factor (HIF-1) and vascular endothelial growth factor (VEGF) pathways [22]. The tumor vasculature can upregulate or downregulate proteins that control the homing and trafficking of immune cells, creating a selective immune-cell barrier on endothelium. The use of anti-angiogenic drugs and other vascular-targeting agents can normalize and remodel the tortuous tumor vasculature, enabling alleviation of hypoxia and efficient tumor infiltration by effector immune cells. Combinations of anti-angiogenic agents with ICIs have been shown preclinically to generate more potential antitumor effects and might have clinical potential [23]. IMmotion151 study (phase III) investigated atezolizumab plus bevacizumab versus sunitinib in patients with previously untreated metastatic renal cell carcinoma. In the PD-L1 positive population, the median progression-free survival was 11.2 months in the atezolizumab plus bevacizumab group versus 7.7 months in the sunitinib group (hazard ratio (HR) 0·74 (95% CI 0·57–0·96); *p* = 0·0217). However, median overall survival had an HR of 0·93 (0·76–1·14), and the results did not cross the significance boundary at the interim analysis [24]. IMpower150 study (phase III) investigated atezolizumab plus bevacizumab plus chemotherapy in patients with metastatic non-squamous non–small-cell lung cancer (NSCLC) who had not previously received chemotherapy. The median progression-free survival was longer in the bevacizumab plus chemotherapy with atezolizumab group than in the bevacizumab plus chemotherapy without atezolizumab group (8.3 months vs. 6.8 months; hazard ratio for disease progression or death, 0.62; 95% confidence interval (CI), 0.52 to 0.74; *p* < 0.001); the corresponding values in the Teff-high WT population were 11.3 months and 6.8 months (hazard ratio, 0.51 (95% CI, 0.38 to 0.68); *p* < 0.001). Median overall survival among the patients in the WT population was longer in the bevacizumab plus chemotherapy with atezolizumab group than in the bevacizumab plus chemotherapy without atezolizumab group (19.2 months vs. 14.7 months; hazard ratio for death, 0.78; 95% CI, 0.64 to 0.96; *p* = 0.02) [25]. All current trials of combination therapy with chemotherapy for cervical cancer are angiogenesis inhibitors and ICI combination therapy. I would like to expect the result (Table 3).

### 4.2. Concurrent Chemoradiation Therapy (CCRT) with ICIs for Cervical Cancer

The combination of radiotherapy and ICIs has been attracting attention. Concurrent chemoradiation therapy (CCRT) is the standard for treating locally advanced cervical cancer. Understanding of the immunogenic potential of radiation has developed over the last several decades through clinical observation, preclinical investigations and, more recently, prospective trails, as discussed herein. Radiation therapy conducts by the principle mechanism of double-strand DNA damage that ultimately leads to cell death. The most well-realized pathway of radiation-induced cellular lethality is mitotic catastrophe, although alternative pathways of cell death exist, including apoptosis, necrosis, autophagy and senescence. All these phenomena lead to a complex interaction between the host immune system and the tumor microenvironment [29]. Activation of antitumor immunity through radiotherapy is called the “abscopal effect”. This effect is a rare but interesting clinical phenomenon whereby tumor regression occurs at a distant metastatic site or sites following local radiotherapy. The abscopal effect was first reported in the clinical literature in 1953 [30]. The incidence of this effect is rare, with a recent systematic review identifying 46 cases in the medical literature, from 1969 to 2014 [31]. Among gynecologic cancers, an abscopal effect has been reported in one patient with local advanced cervical cancer who received pelvic radiation and HDR brachytherapy [32]. Regression of untreated bulky para-aortic nodes was observed.

Preclinical work has shown that the abscopal effect is mediated by cytotoxic T cells, and that radiation can boost the abscopal response by increasing tumor immunogenicity [33]. Radiotherapy to a tumor releases tumor-associated antigens (TAAs) and damaged DNA that can stimulate the production of type-I interferon (IFN-I) via the cyclic GMP–AMP synthase (cGAS)/stimulator of interferon genes (STING) pathway in tumor cells. The subsequent secretion of IFN-I and interferon-stimulated genes (including CXCL10) promote the recruitment and activation of dendritic cells. TAAs are then taken up by antigen-presenting cells (APCs), such as macrophages and dendritic cells. Radiotherapy also results in the release of damage-associated molecular patterns (DAMPs) and pro-inflammatory cytokines, leading to maturation and activation of dendritic cells. Upregulation of PD-L1 and MHC class I on tumor cells also occurs. APCs migrate to lymph nodes to present tumor neoantigens to cytotoxic T cells. Activated T cells induce immunogenic tumor cell death by acting on the primary irradiated tumor, as well as distant metastatic sites [29,33] (Figure 1). However, PD-L1 is upregulated following irradiation and chemotherapy, prompting T-cell apoptosis and anergy upon ligation to PD-1 and restricting the immune response. The addition of anti-PD-1/PD-L1 blocks this interaction, facilitating synergistic antitumor immunity [34].

In prospective clinical studies, the combination of CCRT and an ICI was investigated, demonstrating positive effects [34]. In a phase III trial of lung cancer that treated patients with durvalumab after CCRT, the median PFS was 17.2 months, which is significantly longer than the 5.6 months seen for placebo [35]. However, no conclusions have been reached regarding the optimal timing for this drug, or whether the therapy is better provided in combination or as maintenance therapy. Based on this, clinical trials have been initiated in cervical cancer patients receiving CCRT. Unfortunately, these studies include only three phase II trials and no phase III trials. Inconsistencies also exist between these studies regarding whether the ICI should be administered as maintenance therapy or combination therapy. When these results emerge, CCRT for advanced cervical cancer may be altered, after remaining unchanged in the past for almost two decades (Table 4).

## 5. Conclusions

Cancer immunotherapy is finally playing a more prominent role as a new therapeutic approach for multiple types of cancer. In turn, this has given new hope to patients with relapsed or recurrent cancer. Nonetheless, compared with other types of cancer, very few phase III trials have been performed for cervical cancer, indicating inadequate evidence. However, it attracts attention that combinations of anti-angiogenic agents with ICIs have been shown preclinically to generate more potential antitumor effects and might have clinical potential. On the other hand, through immune system priming, radiation treatment has the potential to enhance systemic response rates to immunotherapy through the abscopal effect, as well as increase local tumor control at the irradiated site. Careful prospective studies will be required to determine the optimal integration of immunotherapy, chemotherapy and radiotherapy. While we await results from currently ongoing trials, we need to narrow down those patients who can expect to see efficacy with good biomarkers and to investigate combination therapy with existing treatments. To understand and develop these issues, further studies on the tumor immune microenvironment and polymorphism analysis of immune-related genes are desired.

## Figures and Tables

**Figure 1 ijms-21-02335-f001:**
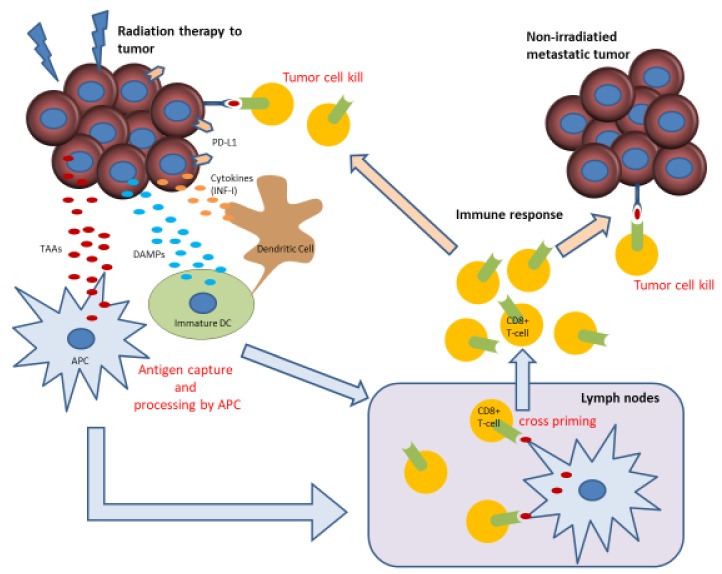
Mechanisms of radiation therapy and immunity (abscopal effect) [29].

**Table 1 ijms-21-02335-t001:** Patient characteristics of KEYNOTE158 and CheckMate 358.

/	KEYNOTE-158 [8]	CheckMate 358 [20]
Treatment	pembrolizumab	nivolumab
Phase	II	I/II
n	98	19
Stage		
II	1 (1.0%)	1 (5.3%)
III	4 (4.1%)	2 (10.5%)
IV	93 (94.9%)	16 (84.2%)
PD-L1 expression		
Positive	82 (83.7%)	10 (62.5%)
Negative	15 (15.3%)	6 (37.5%)
Unknown	1 (1%)	3 (15.8%)
Previous radiotherapy	85 (86.7%)	17 (89.5%)
Previous line of systemic therapy		
0	0 (0%)	0 (0%)
1	30 (30.6%)	8 (42.1%)
2	34 (34.7%)	8 (42.1%)
3	16 (16.3%)	3 (15.8%)
Previous antineoplastic agents		
Cisplatin	79 (80.6%)	15 (78.9%)
Carboplatin	66 (67.3%)	11 (57.9%)
Paclitaxel	85 (86.7%)	12 (63.2%)
Bevacizumab	41 (41.8%)	6 (31.6%)
Topotecan	17 (17.3%)	5 (26.3%)

**Table 2 ijms-21-02335-t002:** Antitumor activity of KEYNOTE158 and CheckMate 358.

/	KEYNOTE-158 [8]	CheckMate 358 [20]
Treatment	pembrolizumab	nivolumab
n	98	19
ORR (95% CI)	12.2% (6.5 to 20.4)	26.3% (9.1 to 51.2)
DCR (95% CI)	30.6% (21.7 to 40.7)	68.4% (43.4 to 87.4)
Best overall response		
CR	3 (3.1%)	3 (15.8%)
PR	9 (9.2%)	2 (10.5%)
SD	18 (18.4%)	8 (42.1%)
PD	55 (56.1%)	6 (31.6%)
Not able to be evaluated*	5 (5.1%)	0 (0%)
Not able to be assessed#	8 (8.2%)	0 (0%)

ORR, objective response rate; DCR, disease control rate; CR, complete response; PR, partial response; SD, stable disease; PD, progressive disease. *Patients who had one or more postbaseline tumor assessment, none of which were evaluable. #Patients who had no postbaseline tumor assessment because of death, withdrawal of consent, loss to follow-up or start of new anticancer therapy.

**Table 3 ijms-21-02335-t003:** Ongoing clinical phase III trials, chemotherapy with immune checkpoint inhibitors (ICIs) for cervical cancer.

Title	Study Population	n	Phase	Treatment	Primary Outcomes	Secondary Outcomes	Clinical Trial Code
Efficacy and Safety of BCD-100 (Anti-PD-1) in Combination with Platinum-Based Chemotherapy with and Without Bevacizumab as First-Line Treatment of Subjects with Advanced Cervical Cancer (FERMATA) [26]	Advanced cervical cancer	316	III	Paclitaxel + cisplatin (or carboplatin) Bevacizumab BCD-100 (anti-PD-1)	OS	PFS, ORR, DOR	NCT03912415
Efficacy and Safety Study of First-Line Treatment with Pembrolizumab (MK-3475) Plus Chemotherapy Versus Placebo Plus Chemotherapy in Women with Persistent, Recurrent, or Metastatic Cervical Cancer (MK-3475-826/KEYNOTE-826) [27]	Recurrent or metastatic cervical cancer	600	III	Paclitaxel + cisplatin (or carboplatin) Bevacizumab Pembrolizumab	PFS, OS	ORR, DCR, DOR	NCT03635567
Platinum Chemotherapy Plus Paclitaxel with Bevacizumab and Atezolizumab in Metastatic Carcinoma of the Cervix [28]	Recurrent or metastatic cervical cancer	404	III	Paclitaxel + cisplatin Bevacizumab Atezolizumab	OS	PFS, ORR, DOR, AE	NCT03556839

OS: overall survival rate; PFS: progression-free survival; ORR: objective response rate; DOR: duration of response; DCR: disease control rate; QOL: quality of life; and AE: adverse event.

**Table 4 ijms-21-02335-t004:** Ongoing clinical phase II trials, CCRT with ICIs for cervical cancer.

Title	Study Population	*n*	Phase	Treatment	Primary Outcomes	Secondary Outcomes	Clinical Trial Code
TSR-042 as Maintenance Therapy for Patients with High-risk Locally Advanced Cervical Cancer After Chemo-radiation (ATOMICC) [36]	Stage IB/IIA/IIB/III/IVA cervical cancer with pelvic or PALN	132	II	CRT Maintenance TSR-042 (anti-PD-1 antibody)	PFS	AE, OS	NCT03833479
Pembrolizumab and Chemoradiation Treatment for Advanced Cervical Cancer [37]	Locally advanced cervical cancer	88	II	Pembrolizumab with CRT	Change in immunologic markers following combination of study drug with chemoradiation, DLT	Metabolic Response Rate on PET/CT imaging, incidence of distant metastases, PFS, OS	NCT02635360
Trial Assessing the Inhibitor of Programmed Cell Death Ligand 1 (PD-L1) Immune Checkpoint Atezolizumab (ATEZOLACC) [38]	Locally advanced cervical cancer	190	II	Atezolizumab with CRT and adjuvant atezolizumab	PFS		NCT03612791

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
