# Peer review of "Immunotherapy for Uterine Cervical Cancer Using Checkpoint Inhibitors: Future Directions"

_ijms, 2020, doi:10.3390/ijms21072335_

Round 1
Reviewer 1 Report
Good review.
Well written and didactic.
Line 87 : various misspelled
Author Response
Thank you for your review and comments.
"Line 87 : various misspelled"
I corrected the misspelling and remake this manuscript.
Reviewer 2 Report
Kagabu et al. reviewed the theoretical background and recent clinical applications of immune checkpoint inhobitors (ICIs). This manuscript is partly based on a previous publication of the authors, reference 7. Kagabu, M.; Nagasawa, T.; Fukagawa, D.; Tomabechi, H.; Sato, S.; Shoji, T.; Baba, T., Immunotherapy or Uterine Cervical Cancer. Healthcare (Basel) 2019, 7, (3). The current review focused, however, mainly on the clinical trials KEYNOTE-158 and CheckMate 358 and included a clear description of the abscopal effect as a potential explanation for the combination of ICIs and radiotherapiy. I have some minor remarks and a major one regarding this manuscript. line 46, ...Agency for Research on Cancer... change to: ...International Agency for Research on Cancer... line 67, ...specific to anti-tumor antigens... change to: ...specific to tumor antigens... line 142, ...CheckMat358... change to: ...CheckMate 358... line 156, ...shown to not necessarily promise an effect... - please rephrase the sentence; line 157, ...Checkmate 358... change to: ...CheckMate 358... line 228 ...I would like to expect the result... - - please rephrase the sentence.
My major concern related to the manuscript is that the authors took over word by word several sentences from their previous publication (reference 7) into the current manuscript. This is certainly not plagiarism in the strict sense of the word but the authors may wish to re-formulate those sentences. E.g. the text in line line 18-23 ...Cervical cancer is a malignant tumor and is the fourth most common cancer among women worldwide. Since the introduction of angiogenesis inhibitors, treatment for recurrent and advanced cervical cancers has improved in the past 5 years, but median overall survival is 16.8 months for advanced cervical cancer and all-stage 5-year overall survival rate is 68%, indicating that treatment effects remain inadequate... corresponds to a part of their Abstract in Ref. 7.
Or, line 287-295 ...High-risk HPV is essential for carcinogenesis and maintaining cancer characteristics for cervical cancer, and immunotherapy targeting HPV has been expected. The E6 and E7 viral proteins, critical in driving HPV oncogenesis and foreign to the human immune system, represent ideal targets for therapeutic cancer vaccination [34]. Many clinical trials of vaccine monotherapy have been conducted. However, the effectiveness of vaccine monotherapy has not been proven for advanced cervical cancer [34]. Clinical trials of vaccine monotherapy have been conducted against CIN3 rather than advanced cervical cancer [34]. In recent years, the development of adjuvants has progressed. Clinical trials of a combination therapy of vaccines for advanced cervical cancer such as
295 ADXS11-001 and ISA101 have been conducted... - is nearly identical with the sentences published in the paragraph 2. Immunotherapy Targeting HPV Related Gene in Cervical Cancer of reference 7 (Kagabu et al., 2019).
I think the text should be double-checked for such overlaps and rephrased where appropriate. I suggest a major revision of the manuscript.
Author Response
Thank you for your review and comments.
This manuscript has been remade to correct minor remarks and major revision.
Minor remarks:
"line 46, ...Agency for Research on Cancer... change to: ...International Agency for Research on Cancer... "
I corrected it.
"line 67, ...specific to anti-tumor antigens... change to: ...specific to tumor antigens... "
I corrected it.
" line 142, ...CheckMat358... change to: ...CheckMate 358... "
I corrected it.
"line 156, ...shown to not necessarily promise an effect... - please rephrase the sentence"
I corrected it.
"line 157, ...Checkmate 358... change to: ...CheckMate 358..."
I corrected it.
"line 228 ...I would like to expect the result... - - please rephrase the sentence."
I corrected it.
"My major concern related to the manuscript is that the authors took over word by word several sentences from their previous publication (reference 7) into the current manuscript. This is certainly not plagiarism in the strict sense of the word but the authors may wish to re-formulate those sentences. E.g. the text in line line 18-23 ...Cervical cancer is a malignant tumor and is the fourth most common cancer among women worldwide. Since the introduction of angiogenesis inhibitors, treatment for recurrent and advanced cervical cancers has improved in the past 5 years, but median overall survival is 16.8 months for advanced cervical cancer and all-stage 5-year overall survival rate is 68%, indicating that treatment effects remain inadequate... corresponds to a part of their Abstract in Ref. 7.
Or, line 287-295 ...High-risk HPV is essential for carcinogenesis and maintaining cancer characteristics for cervical cancer, and immunotherapy targeting HPV has been expected. The E6 and E7 viral proteins, critical in driving HPV oncogenesis and foreign to the human immune system, represent ideal targets for therapeutic cancer vaccination [34]. Many clinical trials of vaccine monotherapy have been conducted. However, the effectiveness of vaccine monotherapy has not been proven for advanced cervical cancer [34]. Clinical trials of vaccine monotherapy have been conducted against CIN3 rather than advanced cervical cancer [34]. In recent years, the development of adjuvants has progressed. Clinical trials of a combination therapy of vaccines for advanced cervical cancer such as
295 ADXS11-001 and ISA101 have been conducted... - is nearly identical with the sentences published in the paragraph 2. Immunotherapy Targeting HPV Related Gene in Cervical Cancer of reference 7 (Kagabu et al., 2019)."
I remade this manuscripts to correct these points.
Round 2
Reviewer 2 Report
The authors correctly addressed the points I made. I suggest the publication of the manuscript.